# Physical Activity, Seasonal Sensitivity and Psychological Well-Being of People of Different Age Groups Living in Extreme Environments

**DOI:** 10.3390/ijerph20031719

**Published:** 2023-01-17

**Authors:** Caren Alvarado, Matías Castillo-Aguilar, Valeska Villegas, Claudia Estrada Goic, Katherine Harris, Patricio Barria, Michele M. Moraes, Thiago T. Mendes, Rosa M. E. Arantes, Pablo Valdés-Badilla, Cristian Núñez-Espinosa

**Affiliations:** 1School of Medicine, Magallanes University, Punta Arenas 6210005, Chile; 2Centro Asistencial de Docencia e Investigación (CADI-UMAG), Punta Arenas 6210005, Chile; 3Kinesiology Department, Magallanes University, Punta Arenas 6210005, Chile; 4Psychology Department, Magallanes University, Punta Arenas 6210005, Chile; 5Corporación de Rehabilitación Club de Leones Cruz del Sur, Punta Arenas 6210005, Chile; 6Brain-Machine Interface Systems Lab, Systems Engineering and Automation Department, Universidad Miguel Hernández de Elche, 03202 Elche, Spain; 7Department of Pathology, Institute of Biological Sciences, Universidade Federal de Minas Gerais, Belo Horizonte 31270-901, MG, Brazil; 8Associate Researcher of the Center for Newborn Screening and Genetics Diagnosis, Faculty of Medicine, Universidade Federal de Minas Gerais, Belo Horizonte 31270-901, MG, Brazil; 9Department of Physical Education, Faculty of Education, Universidade Federal da Bahia, Salvador 40170-110, BA, Brazil; 10Department of Physical Activity Sciences, Faculty of Education Sciences, Universidad Católica del Maule, Talca 3530000, Chile; 11Sports Coach Career, School of Education, Universidad Viña del Mar, Viña del Mar 2520000, Chile; 12Interuniversity Center for Healthy Aging, Santiago 8380544, Chile

**Keywords:** physical activity, seasonal affective disorder, mental health, extreme environments

## Abstract

Physical activity can prevent many organic and mental pathologies. For people living in extreme southern high-latitude environments, weather conditions can affect these activities, altering their psychological well-being and favoring the prevalence of seasonal sensitivity (SS). This study aims to determine the relationships between the practice of physical activity, seasonal sensitivity and well-being in people living in high southern latitudes. A cross-sectional study was conducted, using the Seasonal Pattern Assessment Questionnaire (SPAQ), applying a psychological well-being scale, and determining sports practice according to the recommendations of the World Health Organization (WHO) for the 370 male (n = 209; 55%) and female (n = 173; 45%) participants. The main results indicated that 194 people (52 ± 7.7 years) reported physical activity. High-intensity physical activity practitioners recorded a significantly lower proportion of SS. In terms of psychological well-being, an adverse effect was found between the Seasonal Score Index (SSI) and five subcategories of the Ryff well-being scale. In conclusion, those who perform high-intensity physical activity have a lower SS, and those who have a higher SS have a lower psychological well-being.

## 1. Introduction

In Chile, 81.3% of the population performs a physical activity or sports [1]. Although this figure generically represents the country’s situation, geographical differences may be influential in this type of participation of people. The Magallanes and Chilean Antarctic Region are found in areas of high southern latitude, defined as latitude 50° to the South Pole. This region is considered a geographical area of extreme cold weather that can be an extreme environment for the human body [2]. In this area, only 36.2% engage in physical activity or sport, which may or may not be directly influenced by the geographical conditions and the seasonal cycle of light [3]. Seasonal changes in natural light in these areas can condition the mood of the people living there, leading to what is known as “seasonal sensitivity” (SS). This condition is the sensitivity of individuals to seasonal variations related to the time of exposure to sunlight (increased during the summer and decreased during the winter) [4], leading to a disorder characterized by hypersomnia, increased carbohydrate appetite, weight gain and extreme fatigue [5,6]. Physiologically, these changes can induce disruptions in the circadian rhythm and neuroendocrine dysregulation (e.g., melatonin, which is directly related to sleep disturbances and serotonin, which is related to depressive symptoms) [7,8]. The autonomic dysregulation, consequent variations in vagal tone, and alterations in cardiac regulation in the face of stress can lead to an increased risk of cardiovascular disease [7,8]. These physiological effects, associated with psycho-social adaptation, can affect mental health whereby higher anxiety and depressive symptoms can occur (REF). SS can be a derivative of a psychopathological phenomenon called seasonal affective disorder (SAD), which generates consequences in the individual’s social adaptation and perception of happiness, and is directly related to poor quality of life [5,9,10]. SAD severity includes a subsyndromal-SAD (S-SAD) cyclical form of “winter blues” to severe depression [11].

Physical activity is a non-pharmacological intervention that results in systemic beneficial changes, modulating the neuroendocrine system (REF), which attenuates depressive and anxiety symptoms and is an effective and accessible treatment for SAD and S-SAD [12], as voluntary physical exercise can be an inducer of neurogenesis and neuronal differentiation in certain brain areas, which has been compared to antidepressant drugs [13,14,15]. On the other hand, the benefits of incorporating regular exercise and maintaining a high level of physical activity as part of daily living activities have been extensively studied and are well-known. Some of these are summarized as improved cardiorespiratory fitness, reduced risk of cardiometabolic diseases, improved self-esteem and mood, promotion of social integration, improved management of chronic diseases and many other associated benefits that translate into a better quality of life at different stages of life [16,17]. On the other side, it has been well-studied that sedentary behaviors increase the risk of cardiovascular diseases, diabetes, obesity, and stroke, among others [18,19]. According to the World Health Organization [20], the classification of a physically active person considers that a certain amount of medium- to high-intensity physical activity is met, for example, 150–300 min of medium- to high-intensity aerobic physical activity [20]. While being physically active would improve health status, the intensity of physical exercise should be taken into account, as it has been shown that moderate-intensity training can decrease depressive symptomatology and levels of proinflammatory cytokines such as tumor necrosis factor-alpha (TNF-α) and high-intensity intervallic training (HIIT), which can also decrease depressive symptoms but increase perceived stress and proinflammatory cytokines compared to moderate-intensity [21,22,23].

Physical activity can be performed both outdoors and indoors. The emerging evidence suggests that outdoor exercise promotes a decrease in perceived stress, mediated by the action of the parasympathetic nervous system; also, it may generate increases in vitamin D levels and a reduction in the risk of myopia compared to the same activity performed indoors [24,25,26]. However, outdoor exercise may be hindered by fluctuating climatic conditions worldwide. Cold weather prevails most of the year in high southern latitude areas. At the same time, there are cyclical changes in natural light about the seasons of the year, thus conditioning the life of the region’s inhabitants [27].

Despite being practiced by less than half of the Chilean population living in high southern latitudes, physical activity may be an intervention with a specific relevance for this population if it attenuates the symptoms of SS. Although this has been studied in other continents, the relationship between physical activity, SS and the well-being of people living in these geographical areas has not been studied in Latin America. This study’s main aim is to evaluate the relationships between the practice of physical activity, SS and well-being in middle-aged and older people living in a high southern latitude. We hypothesized that the practice of physical activity presents an inverse correlation with seasonal sensitivity and a positive correlation with the well-being of southern latitude residents; also, the intensity of physical activity may influence this relationship.

## 2. Materials and Methods

### 2.1. Study Design

This study is a non-experimental correlational study. It was conducted in a single stage by applying a presential questionnaire.

### 2.2. Participants

The participants in this study were selected by non-random, accidental sampling from the city of Punta Arenas, Chile, located at latitude 53° south. In total, 370 adults and middle-aged and older people participated in this study. Participants’ gender, age, city of origin, length of stay in the region and presence of psychological illnesses were registered using an anamnesis. The participants were invited to participate mainly through social networks and promotional posters of the research. Length of stay in the high latitude south ALS region was surveyed for each participant.

Inclusion criteria included being of legal age, residing in the city of Punta Arenas for at least six months of the year, not having any degree of disability and being able to read and answer the questionnaire. The exclusion criteria were as follows: if they did not comply with the rules for filling out the form, if they were pregnant, and if they had incomplete answers to any questionnaires. Twelve subjects were excluded from this study. Three hundred and fifty-eight persons were part of the final sample, with 56% male (n = 202) and 44% female (n = 156) participants. The age was grouped into adult subjects (18 to 40 years; n = 200; age 28.2 ± 6.1 years) or middle-aged and older people (older than 40 years; n = 158; age 54.7 ± 10.7 years).

### 2.3. Ethics

Participating subjects gave their permission through informed consent before participation. The Ethics Committee approved this study of the University of Magallanes, Chile (code: Nº045SH2019), following the regulations established by the Declaration of Helsinki on ethical principles in human beings. The volunteers were informed about the research objectives and all the experimental procedures before giving their written informed consent for participation in this study.

### 2.4. Measures

#### 2.4.1. Seasonal Pattern Assessment Questionnaire (SPAQ)

The Seasonal Profile Assessment Questionnaire (SPAQ) is a self-administered and timeless screening tool to access seasonal variation [28,29] experienced in six items: sleep duration, social activity, mood, weight, appetite and energy level. Each item is rated on a five-point scale from “not changing” (0 points) to “changing a lot” (4 points). The sum of six SPAQ items produces an overall Seasonal Score Index (SSI, from 0 to 24 points), with higher scores corresponding to greater sensitivity to seasonal changes. Seasonal Affective Disorder (SAD) reflects a depressive picture with a seasonal pattern (SP) and the winter blues, which is a milder form of SAD, a sub-syndrome (S-SAD) [30]. Furthermore, respondents indicated the degree of severity of seasonal changes from “light” (1 point) to “disabling” (5 points), determining whether seasonal changes are considered a problem.

The analysis of combination of SSI scores with the evaluation degree of severity of seasonal changes indicated the presence of SAD (SSI  ≥  11 and seasonal changes are a problem reported as equal to or greater than moderate ≥ 2) [31] or S-SAD (GSS = 9 or 10 and the seasonal changes scored as equal problem or higher than moderate ≥ 2) [30].

#### 2.4.2. Psychological Well-Being Ryff Scale

Psychological well-being (PWB) 42-item Ryff scale addresses six different dimensions: self-acceptance, a person’s ability to feel good about themselves; positive relationships, a person’s perception of establishing stable, social relationships and having friends that they can trust; autonomy, a person’s ability to resist social pressure to a greater extent and to self-regulate their behavior; environmental mastery, personal ability to choose or create favorable environments to meet one’s needs; personal growth, striving to develop one’s potential and maximize one’s capabilities; purpose in life, which refers to a person’s ability to define a set of goals that enable them to give their life some meaning. Each of these instruments is easy to apply, and in total, four sheets were answered in a Likert-type format for the convenience and speed of the participant [32].

#### 2.4.3. Physical Activity

The report on physical activity engagement was obtained using a survey of selection questions, and the classification of sports subjects was based on the World Health Organization recommendations [19]. According to the World Health Organization, a person is considered physically active when they engage in moderate-intensity physical activity for at least 150–300 min, vigorous-intensity physical activity for 75–150 min, or an equivalent combination of both moderate and vigorous activities throughout the week. Muscle-strengthening activities performed on two or more days a week involving all major muscle groups are also considered [20]. The intensity of the physical activity carried out was self-reported by each participant based on the intrinsic perception that the activity provoked in them. The classification the questionnaire gave was grouped as low-, moderate- and high-intensity.

### 2.5. Procedures

Participants voluntarily signed an informed consent form and then completed the complex instruments in a single session in the following order: physical activity, SPAQ, psychological Ryff well-being scale and demographic data. These were self-administered during the winter, as the assessment of a symptom’s presence is more direct at this time (average daylight hours winter: 2.8; summer: 7.4). Each examination was scheduled in a free time of 1 h to answer all questions.

### 2.6. Statistical Analysis

The data are presented as median (Mdn) and interquartile range (IQR) for continuous variables; for categorical/discrete variables, the absolute and relative sample size was reported.

A non-parametric approach was used since the underlying distribution of measured outcomes, assessed through analytical and graphical methods, did not follow a Gaussian distribution. The Wilcoxon ($W_{Mann-Whintney}$) and Kruskal–Wallis ($\chi_{Kruskal-Wallis}^{2}$) rank sum tests were used for between-subjects analyses, meanwhile the chi-square test ($\chi^{2}$) was used to evaluate goodness-of-fit ($\chi_{gof}^{2}$) and independence of factors ($\chi_{Pearson}^{2}$). In order to assess the association between numeric variables, Spearman’s rho statistic (${\hat{\rho }}_{Spearman}$) was calculated. Effect sizes and their respective 95% confidence intervals are also presented for each statistic.

A probability of committing a type I (α) error of less than 5% (*p* < 0.05) was considered sufficient evidence for statistical significance in hypothesis testing. All the statistical analyses were computed and implemented in the R programming language [33].

## 3. Results

Of the 358 volunteers, 216 (60.3%) were classified as SAD and 55 (15.4%) as S-SAD. One hundred and ninety-four persons (54.2%) reported being engaged in physical activity. With the increasing levels of self-reporting of physical activity intensity, there was a significant decrease in the proportion of people with SAD (Figure 1).

The SSI had a proportional effect on the summer pattern variable. Subjects classified within the SAD group scored higher on the summer pattern than the winter blues and normal groups ($\chi_{Kruskal-Wallis}^{2}$ (2) = 20.76, *p* < 0.001, $\hat{\epsilon }$ = 0.06 and CI95% [0.03, 1.00]). A similar trend was observed for the SSI on the winter pattern, where the SAD group recorded higher scores on the winter pattern than those in the winter blues and normal groups ($\chi_{Kruskal-Wallis}^{2}$ (2) = 52.28, *p* < 0.001, $\hat{\epsilon }$ = 0.15 and CI95% [0.09, 1.00]); those with a mixed-type pattern had a higher proportion of people with SAD compared to those with a winter pattern (Figure 2B). Regarding the self-reported severity of seasonal sensitivity, it was observed to have a positive effect on SSI (Figure 2A). Furthermore, a positive correlation between winter and summer patterns was observed (${\hat{\rho }}_{Spearman}$ = 0.48, CI95% [0.40, 0.56] and *p* < 0.001). On the other hand, when looking at the relationship between seasonal sensitivity and participants’ well-being, a negative correlation was found between the winter pattern and the subcategory of autonomy (${\hat{\rho }}_{Spearman}$ = −0.11, CI95% [−0.21, 0.00] and *p* = 0.044).

In terms of gender, male subjects have a higher score in the environmental control than females ($W_{Mann-Whintney}$ = 18106.5, *p* = 0.01, ${\hat{r}}_{biserial}$ = 0.15 and CI95% [0.04, 0.27]) and, in turn, a higher score in the purpose in life domain ($W_{Mann-Whintney}$ = 18084.5, *p* = 0.01, ${\hat{r}}_{biserial}$ = 0.15 and CI95% [0.03, 0.27]). Although there were no significant differences in relation to the different age groups of the participants, in the total sample, a negative effect was found between SSI and five subcategories of the Ryff well-being scale, namely self-acceptance, autonomy, environmental mastery, personal growth and purpose in life (Figure 3).

## 4. Discussion

The main objective of this study was to determine the relationship between the practice of physical activity, SS and the well-being of people living in high southern latitudes. According to our results, 54.9% of our study population stated that they did some physical activity or sport, a figure higher than the 2017 records in the Magallanes and Chilean Antarctic Region [34]. This may be due to the fact that over these five years, the population that performs physical activity has increased.

In our study population, 76% have some degree of SS, and of this percentage, 87% consider that seasonality is a problem for them. In university students, who circumstantially live in high latitudes, it has been seen that almost half of the subjects studied do not perceive SS as a problem [29]. This difference could be due to the fact that circumstantial exposure to seasonal changes does not allow it to be perceived as a problem, unlike those who live permanently under these environmental conditions.

People who live in Asia and in European areas can also perceive SS; however, it is more limited to specific times of the year, which favors being attentive and aware of the health problems that SS involves [35]. In inhabitants who live in high latitudes, information that the characteristics of the geographical area can influence the problems of this condition in their health, as well as in their state of mind. Based on the above, a direct relationship was found between SS and the classification of the perceived severity of SS (Figure 2), where the higher the perceived severity, the higher the proportion of people with SS. However, a percentage of people still continue to believe that SS is not a problem for them and therefore, do not take action despite the fact that seasonal light affects them.

On the other hand, 84% state that they exercise with a medium –high intensity of physical activity, which is in the healthy ranges compared to the world population [36]. Although some studies report that seasonality affects physical activity regardless of the country in which it is practiced [37] the intensity variable reported by the subjects of this study is inversely related to SS. Thus, it can be considered that the more intense the physical activity, the lower the probability of SS (Figure 2). This may be because physical activity can generate a regulation of the circadian rhythm, generating a lower perception of SS. In the review by Escames et al. [38], it is mentioned that exercise of variable durations and intensity can create changes in the circadian cycle regardless of the time of exposure to light. On the other hand, one of the known benefits of exercise is the improvement of mood disorders, which may be another reason that there is an association between the intensity of the activity and the perception of SS [39,40].

Regarding the perception of psychological well-being, an inverse relationship was found between SS and multiple domains of the Ryff Scale: self-acceptance, autonomy, control of the environment, personal growth and life goals. The presence of SS negatively impacts the subject’s psychological well-being, which can lead to a decrease in the performance of physical and sport activities, further increasing the probability of suffering from SS to some degree.

This cyclical relationship between psychological well-being, seasonal sensitivity and exercise is a determining factor, taking into account that the study area is one of the southernmost in the world with a high rate of tourism and, therefore, with a greater flow of people, since seasonal light changes can generate variations in their previous psychological well-being [41]. It is important to raise awareness about the benefits of physical activity in these settings, to avoid or reduce the impact of SS on psychological well-being. In a study carried out in South Africa, where seasonal changes are not so drastic, regular exercise was associated with significant improvements in the total well-being score and especially in the well-being components of mood and sense of coherence, which demonstrates the importance of these practices in the general population [41].

One of the limitations of this study is that it was impossible to control the type of physical activity performed by the participants, which could generate a degree of error in categorizing the type of exercise performed. Another important factor to consider in future studies is the objectification of the duration and intensity of the physical activity performed. Furthermore, we believe that future research should include physiological measurements to determine the impact of SS on biological domains.

## 5. Conclusions

In this study, it was possible to determine that people who perform more intense physical activity may have lower SS, enabling greater well-being. On the other hand, SS can affect psychological well-being in multiple domains, which implies a risk to the mental health of those who present it. However, many people with SS do not perceive it as a problem, which could exacerbate the condition.

Based on the results of this study, sports programs can be designed taking into account the reality of these regions and thus, promote physical activity and sport, favoring the health of the inhabitants with absolute relevance to their development.

## Figures and Tables

**Figure 1 ijerph-20-01719-f001:**
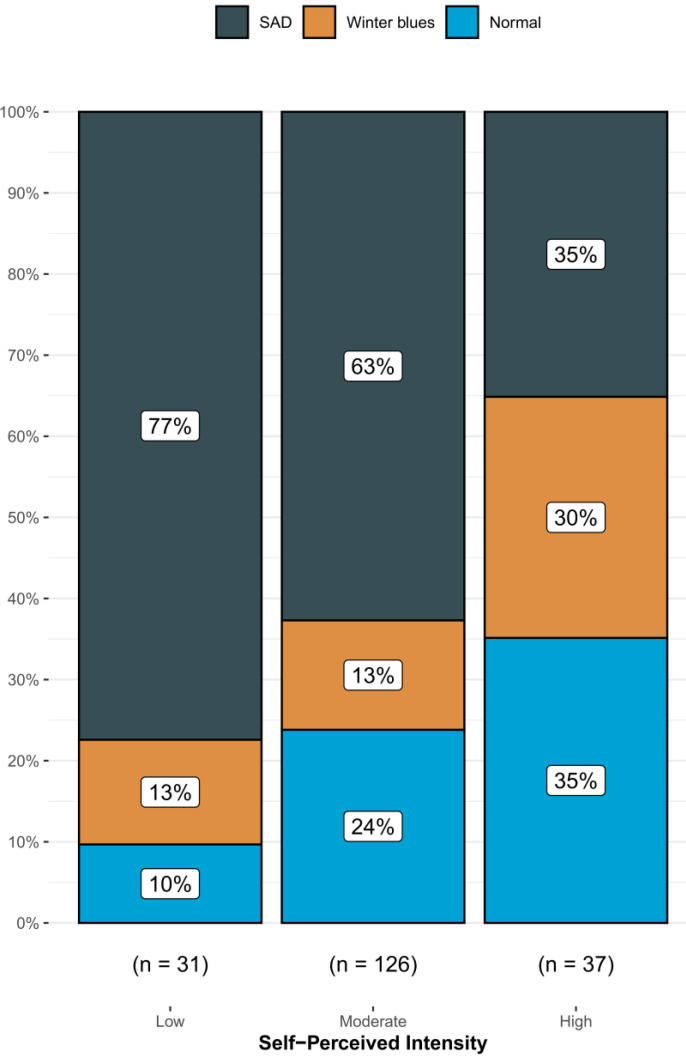
Cumulative bar plots indicate the observed proportions between the levels of SSI and the self-perceived intensity of physical activity.

**Figure 2 ijerph-20-01719-f002:**
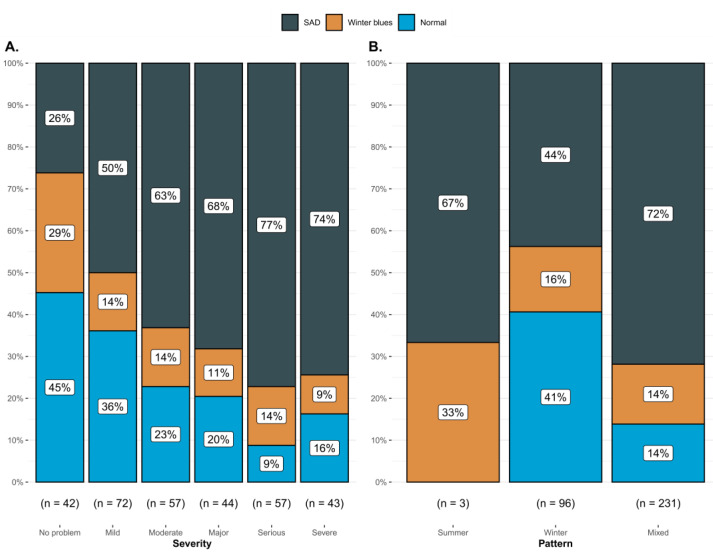
Cumulative bar plots indicating observed proportions between different levels of SSI and (**A**) the severity of self-perceived seasonality, (**B**) and the proportions between the type of seasonal pattern.

**Figure 3 ijerph-20-01719-f003:**
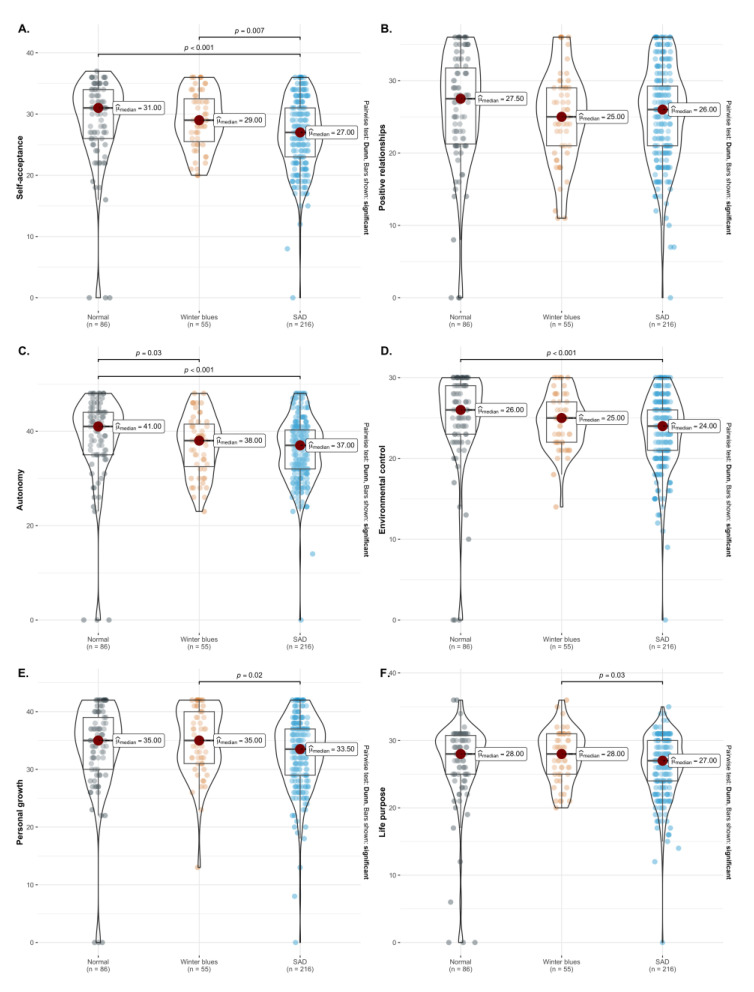
Violin plots indicate the observed differences between different SSI levels on Ryff’s parameters of psychological well-being. Dunn’s test was applied and highlighted for every significative pairwise difference for each panel. *p*-values are shown unadjusted for multiple comparisons (given the study’s exploratory nature).

## Data Availability

The datasets generated during and/or analyzed during the current research is available from the Corresponding author upon reasonable request.

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
