# Peer review of "Physical Activity, Seasonal Sensitivity and Psychological Well-Being of People of Different Age Groups Living in Extreme Environments"

_ijerph, 2023, doi:10.3390/ijerph20031719_

Round 1

Reviewer 1 Report

The title is unclear about the "extreme environments". The study did not clearly define the cut-off of low, medium and high level of physical activity, it makes the results inconclusive. The demography of subjects was not well-presented such as age, gender, education background. These factors are linking to the well-being and physical activity, but not discussed in the study. Apologize that the study needs an extensive revision on the data analysis, result presentation, discussion and conclusion.

Author Response

Dear reviewer,

We thank each one of the reviewers, the time and work of having reviewed our work.

We have improved the paper in its entirety, including the comments of the three reviewers.

We attach the response to each of the suggestions made in the format proposed by the journal.

Reviewer 2 Report

We thank you for thinking of our magazine to send your work. The article is interesting although there is already a lot written on this topic. However, I understand that work continues in this regard to improve situations that can occur in different subjects. My advice is that not so many authors sign this type of work because the merit is lost by being divided among so many authors.

Author Response

Dear Reviewer 2:

We thank each one of the reviewers, the time and work of having reviewed our work.

We have improved the paper in its entirety, including the comments of the three reviewers.

We attach the response to each of the suggestions made in the format proposed by the journal.

Reviewer 3 Report

Dear Authors,

The theme of your study is extremely important today. The issue of mental health is something that has gained particular prominence, especially after the pandemic. Therefore, this should be highlighted in your manuscript in some way. This approach would help justify and frame the relevance of your analysis. 

Other considerations that should be made in order to improve the publication potential of your article are related to the following aspects:

1. The abstract, although complete, does not contextualize the empirical study. We only get this notion towards the end of the Introduction. This information should be revealed from the very first moment. You may also consider including this information in a subheading. 

2. The literature review on the relationship between mental health and physical exercise should be further explored and extended (only 36 references are given in the whole article) in order to provide more robustness to the study. 

3. Also the discussion of the results should also be more thorough, establishing comparisons with previous studies carried out in similar geographical contexts. 

4. The conclusion should also be more developed, indicating the limitations of the study and presenting future research proposals. 

I hope this review can contribute to an improvement of your work. 

Best regards,

The Reviewer

Author Response

Dear Reviewer 3:

We thank each one of the reviewers, the time and work of having reviewed our work.

We have improved the paper in its entirety, including the comments of the three reviewers.

We attach the response to each of the suggestions made in the format proposed by the journal.

Round 2

Reviewer 1 Report

Congratulation to the great improvement in the manuscript. The topic is precise and concise that it will attract follow research to explore the human response in extreme environments. Understand that the authors may not be native English speakers (So do I), the use of terminology and expression has a room for improvement. Basically, appreciated the effort made by the authors to revise the manuscript.

Author Response

Dear reviewer,
We appreciate your comments once again.
We have made the suggested changes. The text presents improvements in terminology and expressions in English.
We appreciate your time and contribution, which undoubtedly helps to improve our work.